# Continuous Electron Beam Post-Treatment of EBF³-Fabricated Ti–6Al–4V Parts

**Alexey Panin** [1,2,*] , **Marina Kazachenok** [1], **Olga Perevalova** [1], **Sergey Martynov** [1], **Alexandra Panina** [2] **and Elena Sklyarova** [2]

[1]  Institute of Strength Physics and Materials Science of Siberian Branch of Russian Academy of Sciences, 634055 Tomsk, Russia; kms@ispms.tsc.ru (M.K.); perevalova52@mail.ru (O.P.); martynov@ispms.tsc.ru (S.M.)

[2]  School of Nuclear Science & Engineering, National Research Tomsk Polytechnic University, 634050 Tomsk, Russia; sonaa@tpu.ru (A.P.); skea@tpu.ru (E.S.)

*  Correspondence: pav@ispms.tsc.ru; Tel.: +7-3822-286-979

**Abstract:** In the present study, the methods of optical, scanning electron, and transmission electron microscopy as well as X-ray diffraction analysis gained insights into the mechanisms of surface finish and microstructure formation of Ti–6Al–4V parts during an EBF³-process. It was found that the slip band propagation within the outermost surface layer provided dissipation of the stored strain energy associated with martensitic transformations. The latter caused the lath fragmentation as well as precipitation of nanosized β grains and an orthorhombic martensite α″ phase at the secondary α lath boundaries of as-built Ti–6Al–4V parts. The effect of continuous electron beam post-treatment on the surface finish, microstructure, and mechanical properties of EBF³-fabricated Ti–6Al–4V parts was revealed. The brittle outermost surface layer of the EBF³-fabricated samples was melted upon the treatment, resulting in the formation of equiaxial prior β grains of 20 to 30 μm in size with the fragmented acicular α′ phase. Electron-beam irradiation induced transformations within the 70 μm thick molten surface layer and 500 μm thick heat affected zone significantly increased the Vickers microhardness and tensile strength of the EBF³-fabricated Ti–6Al–4V samples.

**Keywords:** additive manufacturing; electron beam free-form fabrication; continuous electron beam post-treatment; surface finish; microstructure; surface hardening

## 1. Introduction

Electron beam free-form fabrication (EBF³) is one of the various additive manufacturing techniques, which has been the subject of keen interest in recent years [1–3]. This process offers the potential for low-cost building of Ti–6Al–4V titanium alloy parts, in particular, which are widely used in aircraft, chemical, medical, and other industries due to their perfect strength to weight ratio, high corrosion resistance, and fracture toughness. The EBF³ process can use commercially available titanium alloy wire of different size and grades, while the process is nearly 100% efficient in feedstock consumption. Moreover, the build rate of the EBF³ process is extremely high (up to 2500 cm³/h) [2]. On the other hand, EBF³-fabricated Ti–6Al–4V parts show a much more coarse grain size feature in comparison to the fine grain size in the parts produced by selective laser melting (SLM) or electron beam melting (EBM) due to the re-melting and cyclic heat treatment of underlying layers during the building up of new layers. Although the cyclic heating is also true for SLM and EBM parts, the width and depth of the molten pool formed in Ti–6Al–4V parts during the SLM process at a 50 W laser power and 0.1 m/s scanning speed (i.e., typical SLM processing parameters) do not exceed 160 and 50 μm, correspondingly [4]. In turn, the width of the molten pool formed in Ti–6Al–4V parts during selective electron beam melting (electron beam power ranges from 160 to 640 W and the scanning speed is equal to 1.6 m/s) varies

from 100 to 500 μm while their depth varies from 100 to 180 μm [5]. However, the typical width of the molten pool formed in the EBF$^3$-fabricated Ti–6Al–4V parts reaches 3.2 mm [6]. This is in a good agreement with our macroscopic simulations of the temperature distribution in the EBF$^3$ process. Moreover, the simulated depth of the molten pool exceeds 3 mm, i.e., 20 times more than the molten pool depth of EBM parts.

In addition, steps are formed at the work piece surface during layer-by-layer deposition (so-called stair stepping effect [7]). Due to a high value of stress concentration factors, the stair-stepping causes a primary source of crack initiation and growth followed by rapid brittle fracture of quite tough (α + β) titanium alloys. To reach the requirements of engineering applications, different surface post-treatment technologies are developed instead of conventional grinding and cutting processes, improving the geometrical stability and surface quality of 3D-printed Ti–6Al–4V parts.

Nowadays, increasing interest is given to ball burnishing [8], shot peening [9], cavitation peening [10], and other mechanical post-treatment technologies that allow not only smoothening of the rough surface finish of as-built parts, but also an increase of their hardness, strength, wear resistance, and fatigue life as well as a reduction their friction due to impact at the surface. Different types of lasers (such as $CO_2$ gas laser, Nd:YAG lasers, copper vapor lasers, and excimer lasers) widely used for engraving, drilling, cutting, and heat treatment of the enormous variety of structural materials can also be successfully adopted for surface finish improvement of additive manufactured metal parts [11]. Besides, the methods of continuous electron beam and pulsed current electron beam treatments can be effectively used as thermal post-processing techniques of 3D–printed titanium components [12]. In particular, simultaneous radiation, heat, and impact treatment of the surface of titanium alloys, which is accompanied by ultrahigh rates of heating (up to temperatures exceeding the melting point) and cooling, provide a so-called self-quenching effect associated with β → α′ transformation via the diffusionless martensitic mechanism that results in the formation of a lamellar martensitic structure. Moreover, the redistribution of alloying elements can occur, leading to the homogenization of the phase composition of surface layers or emergence of new metastable phases and compounds which cannot be obtained using traditional methods of heat treatment.

As compared with pulsed electron beam processing [13], continuous electron beam scanning treatment [14] is capable of increasing the strength of 3D-printed parts with a large surface area. Using a two-dimensional beam deflection technique, the surface morphology and microstructure of the irradiated surface layer can be tailored over a wide range by changing both the electron beam power and the speed of the sample movement during the treatment. Moreover, since a deflection scanning system is equipped on an EBF$^3$ machine, it is easy to combine electron-beam additive manufacturing with continuous electron beam post-processing. The aim of this work is to study the effect of continuous electron beam post-treatment on the surface finish, microstructure, and mechanical properties of EBF$^3$-fabricated Ti–6Al–4V titanium alloy parts.

## 2. Materials and Methods

EBF$^3$ was performed using an electron beam welding machine, 6E400 (Teta, Tomsk, Russia), operating in a vacuum below $1.3 \times 10^{-3}$ Pa at an acceleration voltage of 30 kV. During the layer-additive process, a commercially available Titanium Grade 5 (Ti–6Al–4V) wire 1.6 mm in diameter was melted with a plasma power source onto the surface of a water-cooled titanium baseplate to build up a part. The distance between the electron beam gun and the 150 mm × 150 mm × 10 mm titanium baseplate (working distance) was 630 mm. The beam current was gradually decreased from 24 to 17 mA during the EBF$^3$ process. The wire was front-fed at an angle of 35° with the baseplate surface. The wire speed was 2 m/min. In total, 100 layers, 85 mm long, 5 mm wide, and 0.7 mm high, were deposited on the baseplate with a travel speed of 5 mm/s.

The electron beam welding machine was also used for the continuous electron beam post-treatment of Ti–6Al–4V parts manufactured by the EBF$^3$ process. Figure 1 depicts the configuration of the experimental setup for the electron beam post-treatment. A sawtooth signal with a frequency of

400 Hz was used to turn the electron beam into a line of a 27 mm length. The accelerating voltage and beam current were 30 kV and 60 mA, respectively. The samples were processed, being moved with a speed of 20 mm/s with respect to the deflected electron beam. The focused electron beam with a spot of 0.5 mm in diameter treated the sample surface line by line. The electron gun power and the sample's movement speed were chosen to provide an energy density of 450 J/cm$^2$. During the treatment, the samples were clamped on a water-cooled table.

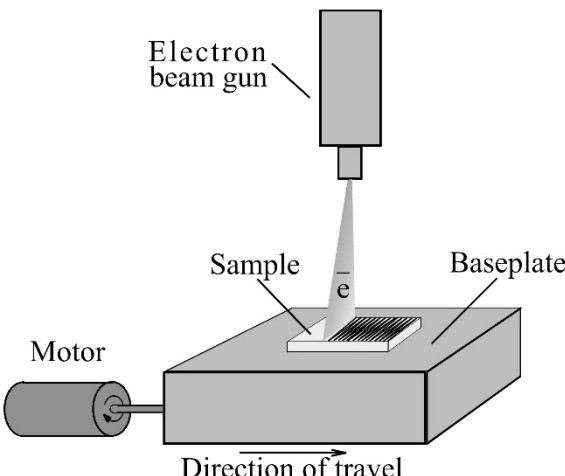

**Figure 1.** Experimental setup for electron beam post-treatment of the surface of Ti–6Al–4V parts.

The surface roughness of Ti–6Al–4V parts in the as-built state and after continuous electron beam post-treatment was measured by a New View 6200 3D optical profiler. A Carl Zeiss Axiovert 40 MAT optical microscope (Carl Zeiss, Göttingen, Germany), a JEM-2100 transmission electron microscope (TEM, JEOL, Tokyo, Japan), and a LEO EVO 50 scanning electron microscope (SEM, Carl Zeiss, Oberkochen, Germany) equipped with EDS detectors were employed for microstructural characterization of the samples in the plan-view and cross-section geometries. TEM samples were prepared by ion milling using Ion Slicer EM-09100IS (JEOL, Tokyo, Japan). During preparation, argon was used as the working gas, the accelerating voltage was 8 kV, and the etching angle was 1.5° to 4°. The ion milling process was controlled by a charged coupled device (CCD) camera. Samples for metallographic study were cut parallel and perpendicular to the build direction using electrical discharge machining. The samples were etched out with Kroll's reagent to reveal their structure after mechanical polishing.

The phase composition of the EBF$^3$-fabricated Ti–6Al–4V parts was determined using a Shimadzu XRD-7000 X-ray diffractometer (XRD, Shimadzu Corporation, Kyoto, Japan). XRD studies were performed with CuK$_\alpha$ radiation (1.5410 Å wavelength) in conventional symmetric Bragg–Brentano geometry from 25° to 80° with a scan speed of 1.2°/min. Residual stress measurements were done using the sin$^2\psi$ [15]. To perform an in-depth study of the microstructure and phase composition of Ti–6Al–4V parts, an X-ray analysis was also done after the removal of a 1 mm thick surface layer by mechanical polishing with emery paper up to grade 1000.

Uniaxial quasi-static tension tests were conducted in air at room temperature using an INSTRON 5582 testing machine (Instron Deutschland GmbH, Darmstadt, Germany) operated at a loading rate of 0.3 mm/min. Mechanical tests were performed with dumb-bell samples cut out of the XY plane (perpendicular to the build direction) by the electrical discharge machining. The gage section of the samples was 1 mm high, 5 mm wide, and 40 mm long. To avoid the effect of surface finish on the mechanical properties of the samples, one part of the samples was mechanically polished before loading.

The Vickers microhardness was measured at the lateral surface of Ti–6Al–4V alloy samples using the 50 g load for 10 s. The first measurement of microhardness was made at a distance of 10 μm

from the top surface. Each microhardness point was obtained by doing parallel measurements at the same distance on the lateral surface and calculating the average value. The distance between any two neighboring indentations was more than 10 μm.

## 3. Results

### 3.1. Surface Finish

The image of a single-pass multi-layer Ti–6Al–4V wall part is presented in Figure 2a. The as-built part is characterized by the surface finish in the form of horizontal layer bands. The layer band morphology along the horizontal direction is formed by consecutive stair steps, which is governed by the flow direction of the molten pool during the layer additive process. The latter is clearly demonstrated by the different morphology of adjacent layers deposited in alternating deposition directions (Figure 2b). The average thickness of alternating layers is equal to 700 μm. Moreover, the peak–trough profile roughness oriented along the build direction is the inherent attribute of the EBF$^3$-fabricated Ti–6Al–4V parts, which is due to surface tension in the molten pool. The quantitative analysis of the as-build Ti–6Al–4V parts carried out by optical profilometry revealed that the height differences between the peak and valley of the profiles directed parallel and perpendicular to the build direction are equal to 150 and 15 μm, correspondingly (Figure 3b, profile 1 and Figure 3c, profile 3). Finally, SEM imaging of the top polished surface and side surface of the wall part evidences that the layers obtained are structurally homogeneous without the presence of pores or cracks.

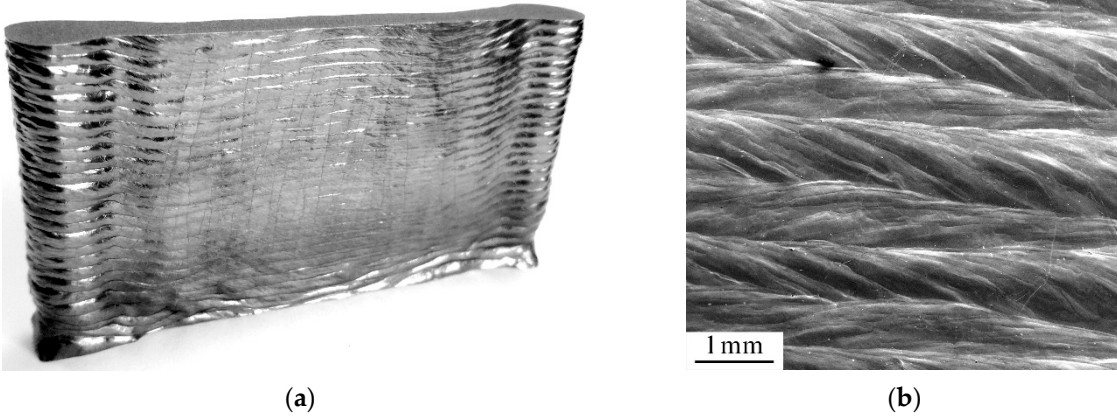

| (a) | (b) |

**Figure 2.** Image (**a**) and surface profile (**b**) of a Ti–6Al–4V sample deposited by the Electron beam free-form fabrication EBF$^3$ process.

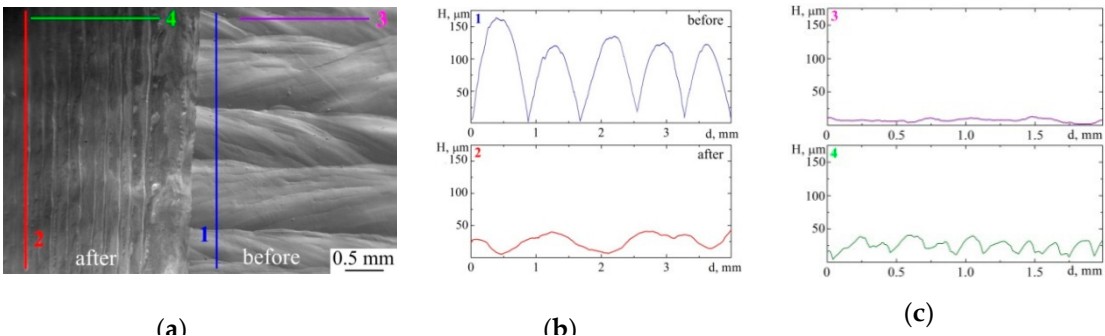

| (a) | (b) | (c) |

**Figure 3.** SEM image (**a**) and corresponding profiles directed parallel (**b**) and perpendicular to the build direction (**c**) of the EBF$^3$-fabricated Ti–6Al–4V sample before (profiles 1, 3) and after scanning electron beam treatment (profiles 2, 4).

Continuous electron beam treatment affords us the opportunity to smooth the surface finish of EBF$^3$-fabricated Ti–6Al–4V parts. During the treatment, high energy electrons are focused to a narrow

spot and deflected so as to form a deep and narrow melted zone on the sample surface. The chosen processing parameters are used to cover the entire surface by multiple overlapping tracks. As a result, a grooved surface topography caused by the pile-up of molten metal near the edge of the melted zone is observed (Figure 3a). Nevertheless, the surface of the Ti–6Al–4V sample becomes smoother after the treatment as compared with the surface finish of the as-built samples. The evidence of the latter is a less rough profile oriented parallel to the build direction: Profile 2 shows a decrease in the average peak to valley height down to 50 μm (Figure 3b). However, due to the appearance of smaller grooves perpendicular to the build direction during the treatment, the difference between the highest peak and the deepest valley of the perpendicular path profile increases up to 30 μm (Figure 3c, profile 4).

### 3.2. Microstructure

The microstructure of the EBF³-fabricated Ti–6Al–4V part in the as-built state consists of the usual columnar prior β grains transformed into a lamellar α-morphology. The prior β grains are epitaxial and extend along the build direction since the heat flux is directed toward the baseplate. The columnar prior β grains reach several millimeters in length that drastically exceeds the alternating layer thickness, while their lateral size varies between 500 μm and 1 mm (Figure 4). A lamellar structure is formed within prior β grains during the cyclic reheating of the solidified metal close to the melt pool. Figure 5c evidences that the lamellar structure consists of an intimate mixture of the lamellae and acicular α′ martensitic phases of different size scales. This trend was also captured by [16], where the martensitic hierarchical structure consisted of primary, secondary, tertiary, and quartic α′ phases formed in SLM Ti–6Al–4V parts. Note that the formation of relatively equiaxial grains in the first deposited layers of the EBF³-fabricated Ti–6Al–4V wall partly due to the intense heat extraction into the baseplate is not under consideration in the present study.

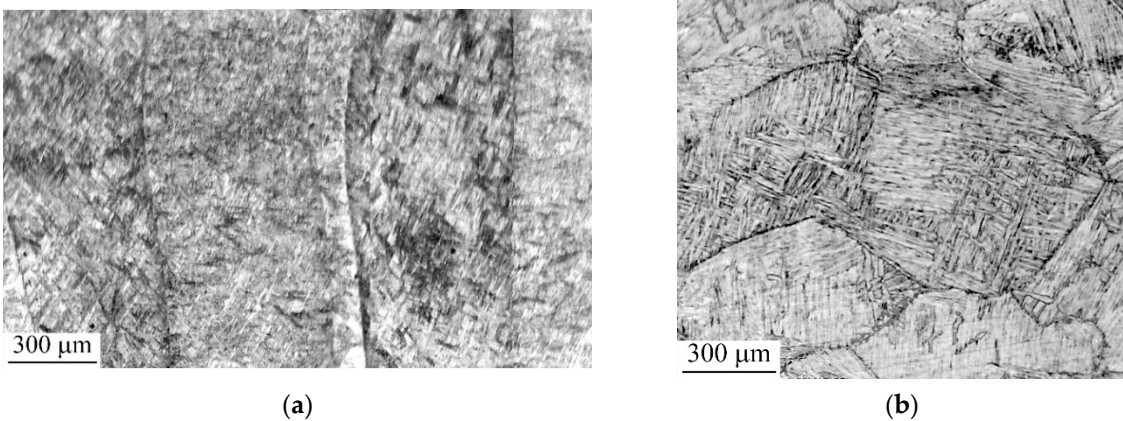

(**a**)　　　　　　　　　　　　　(**b**)

**Figure 4.** Grain microstructure of the EBF³-fabricated Ti–6Al–4V sample in the longitudinal (**a**) and transversal directions (**b**).

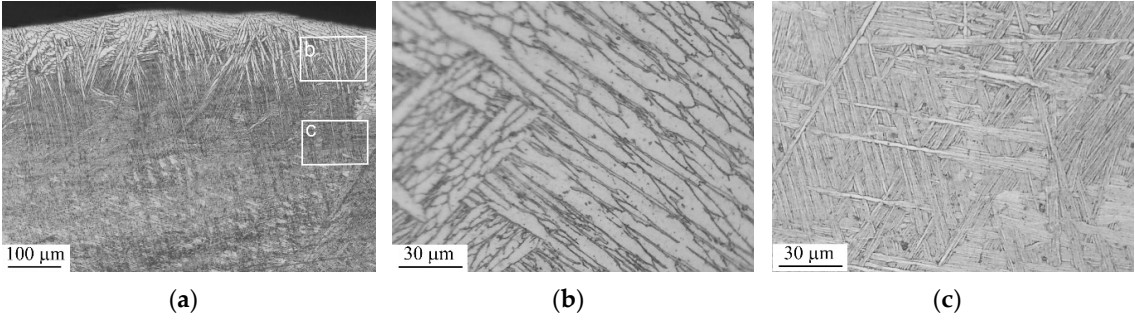

(**a**)　　　　　　　　(**b**)　　　　　　　　(**c**)

**Figure 5.** Microstructure within the outermost surface area (**a**,**b**) and the core region (**c**) of the EBF³-fabricated Ti–6Al–4V sample in the as-built state.

Indeed, inhomogeneous crystallization of building layers is a noticeable lacking from the wire-feed EBF³ process. Less rapid heat transfer causes a slower solidification and cooling rates of the surface area compared with the core area of the Ti–6Al–4V parts during the layer additive process. Since the length and width of α-Ti plates, colonies, laths, etc. mainly increase with a decreasing cooling rate [17,18], it is not surprising that the microstructure of the approximately 100 μm thick outermost surface layer of the samples under study is composed of pockets of massive martensite, which consists of coarse α lathes (100 μm in length and 5 to 10 μm in thickness) (Figure 5a,b). TEM cross-sectional study affords us the opportunity to provide the evidence of secondary martensite transformation within the outermost surface layer (Figure 6).

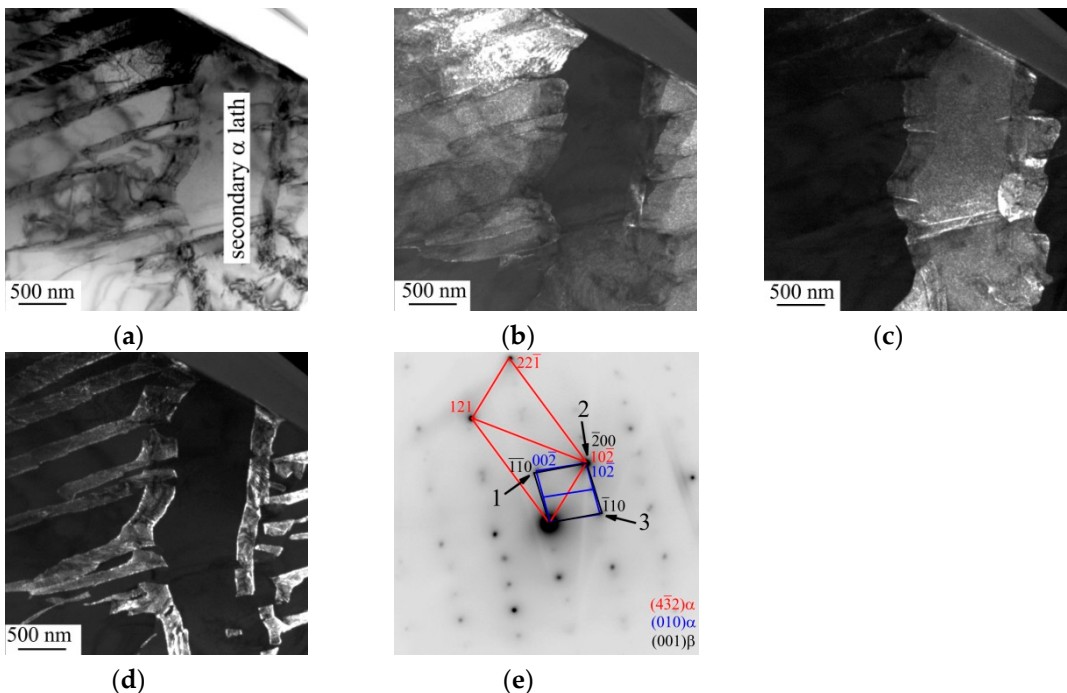

**Figure 6.** TEM bright- (**a**) and dark-field images (**b**–**d**) and associated selected area electron diffraction (SAED) pattern (**e**) of the microstructures within the outermost surface layer of the EBF³-fabricated Ti–6Al–4V sample. b—the dark-field TEM image obtained with the $00\bar{2}_{\alpha\text{-Ti}}$ and $\overline{1}10_{\beta\text{-Ti}}$ reflections (marked by an arrow 1 in (**e**)), c—dark-field TEM image obtained with the $10\bar{2}_{\alpha\text{-Ti}}$ and $\bar{2}00_{\beta\text{-Ti}}$ reflections (marked by an arrow 2 in (**e**)), d—dark-field TEM image obtained with the $\overline{1}10_{\beta\text{-Ti}}$ and $002_{\alpha''\text{-Ti}}$ reflections (marked by an arrow 3 in (**e**)).

Martensitic transformations are known to be responsible for an increase in stored strain energy in order to accommodate an associated volume change [19]. The dissipation of the stored strain energy within the outermost surface layer occurs by slip band propagation. Kinks on the secondary α lath boundary clearly demonstrate the mutual displacement within the primary α lath over a distance of 200 to 300 nm (Figure 6b). Besides, the dislocation sliding gives rise to fragmentation of the primary and secondary martensitic lathes and to torque in the regions surrounding the slip bands. The later manifests itself as misorientations of the matrix between neighboring slip bands, which are by no means negligible. Shear-driven grain structure evolution in SLM Ti–6Al–4V parts during the high-rate cooling from temperatures above the β transus temperature is discussed in detail in [20].

Moreover, it is found that secondary martensite transformation during layer-by-layer deposition provides precipitation of the β and α″ phase in the high-angle boundary of secondary α laths (Figure 6d). To reveal the precipitation mechanism of β and α″ phases in the high-angle boundaries within the outermost surface layer of as-built Ti–6Al–4V parts, a TEM sample was intentionally tilted using a goniometer stage. Dark-field TEM images (Figure 6b–d) evidence the same width (approximately

200 nm) of the projection of the titled high- and low-angle boundaries in the image plane. Although it is hard to identify which edge of the projection belongs to the upper surface and which belongs to the lower surface of the sample in this case, the observation of globular nanoparticles of the β phase and thin platelets of α″ martensite at the projected secondary α lath boundaries is recognizeable in Figure 6d.

High-magnification TEM images clearly demonstrate the regularly arranged platelets of α″ martensite (Figure 7). The transverse dimensions of the α″-phase platelets reach 5 nm while their longitudinal dimensions are limited by the high-angle boundary of secondary α laths.

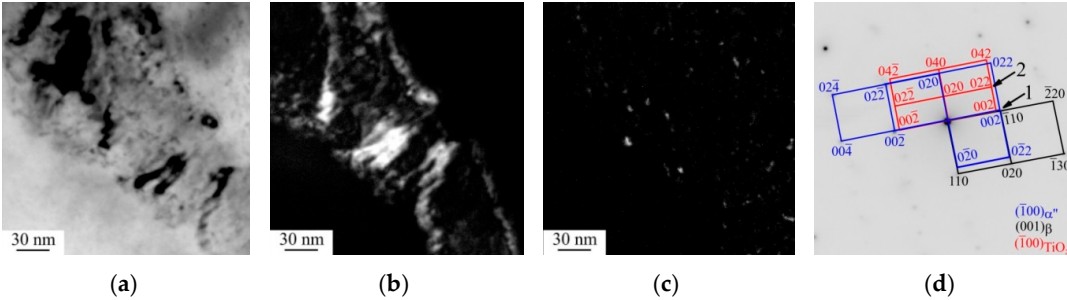

(**a**)                (**b**)                (**c**)                (**d**)

**Figure 7.** TEM bright- (**a**) and dark-field images (**b**,**c**) and associated selected area electron diffraction (SAED) pattern (**d**) of the microstructures within the outermost surface layer of the EBF³-fabricated Ti–6Al–4V sample. **b**—the dark-field TEM image obtained with the $002_{\alpha''\text{-Ti}}$, $\bar{1}10_{\beta\text{-Ti}}$, and $002_{\text{TiO}_2}$ reflections (marked by an arrow 1 in (**d**)), **c**—dark-field TEM image obtained with the $022_{\text{TiO}_2}$ reflection (marked by an arrow 2 in (**d**)).

Due to the extremely high cooling rate of Ti–6Al–4V parts produced by additive manufacturing [21], α′ martensite phase is typically highly enriched with vanadium, which is responsible for a small amount of retained β phase. During re-melting and cyclic heat treatment of the solidified metal close to the melt pool, the decomposition of the supersaturated solid solution α′ phase and the precipitation of β phase in high-angle boundary of secondary α lathes occurs. Herein, dislocation sliding partly promotes the martensitic transition of a highly strained metastable β-Ti crystal lattice into the lattice of the orthorhombic martensite α″ phase. Since the volume of the unit cell of the α″ phase ($0.06998$ $\text{nm}^3$) is twice as high as that of the β phase ($0.03659$ $\text{nm}^3$), it is intensive shear-induced slip bands accompanied by tension that make possible the β → α″ phase transformation in the high-angle boundary of secondary α laths. The appearance of orthorhombic α″ martensite phase in 3D printed Ti–6Al–4V alloy samples produced by electron beam melting [22] or selective laser melting [23] is typically explained by the concentration of β-stabilizer in the β phase. Nevertheless, the appearance of strain-induced α″ phase from the β phase has been observed in Ti–6Al–4V alloy failed by adiabatic shear [24] or in some β-metastable titanium alloys (e.g., TiNbTaZr alloy [25], TiAlMoVCr [26]) subjected to less severe deformation (e.g., cold rolling or compression test).

STEM and TEM cross-section images of the massive martensite structure in the core of EBF³-fabricated Ti–6Al–4V parts are shown in Figures 8 and 9. Massive martensite consisting of large irregular zones (5–10 μm in size) divided across into approximately 2 μm thick α-lathes are revealed in the STEM image (Figure 8a). Light-field and dark-field TEM images taken at a higher magnification display that the 2 μm thick α-lathes consist of the 200 to 300 nm thick α-lathes separated by the retained β phase (Figure 9).

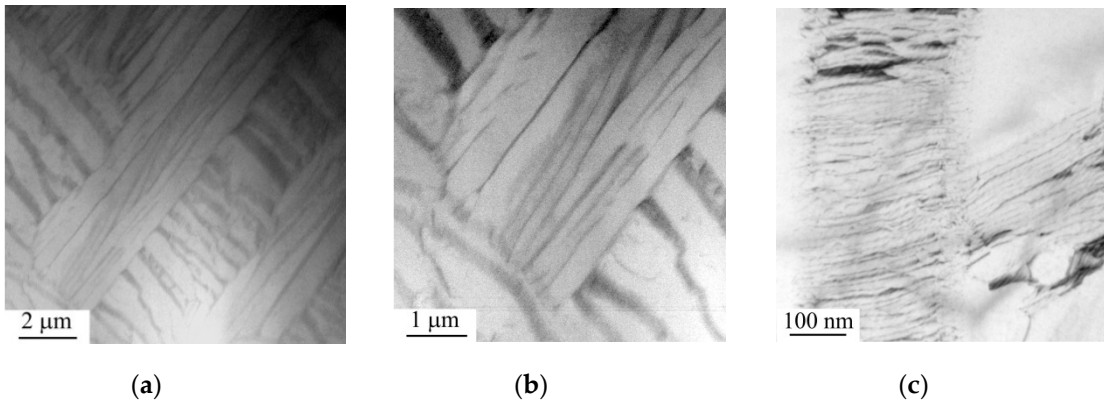

**Figure 8.** STEM (**a,b**) and TEM bright-field images (**c**) of the microstructures of the core of the EBF³-fabricated Ti–6Al–4V sample.

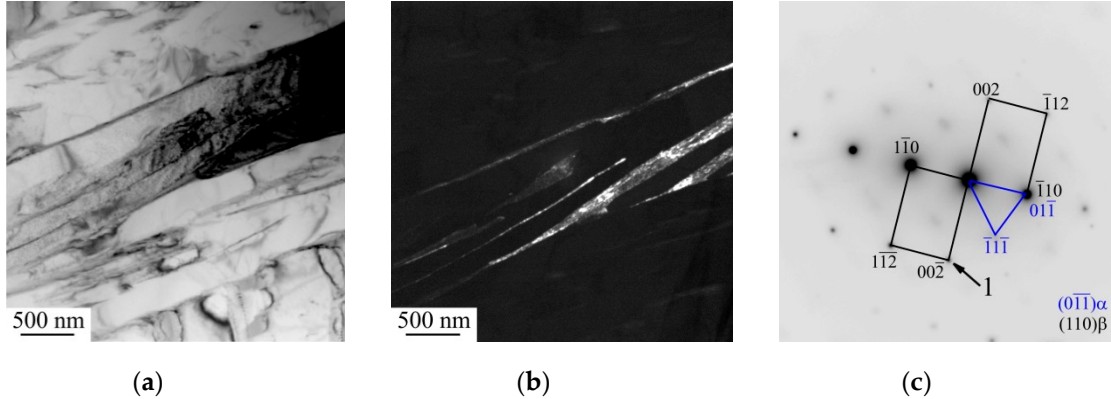

**Figure 9.** TEM bright- (**a**) and dark-field images (**b**) and associated SAED pattern (**c**) of the microstructures of the core of the EBF³-fabricated Ti–6Al–4V sample. The dark-field TEM image obtained with the $00\bar{2}_{\beta\text{-Ti}}$ reflection (marked by an arrow 1 in (**c**)).

Similar to slip band propagation within the outermost surface layer, the formation of low-angle boundaries (Figure 8c) and twins (Figure 10) effectively accommodate martensite transformation in the core of Ti–6Al–4V parts. Although the slip is almost always dominant due to the fact that the stress required is far less than the twinning stress, the deformation twinning in the core of the EBF³-fabricated sample is more likely than in the outermost surface layer due to the faster cooling rate of the core region. It is also evident from Figure 8b,c that the remained dislocations provide the preferential nucleation sites for martensite to enhance the nucleation rate, hence, resulting in secondary α lathes [27] during α → β → α transformation.

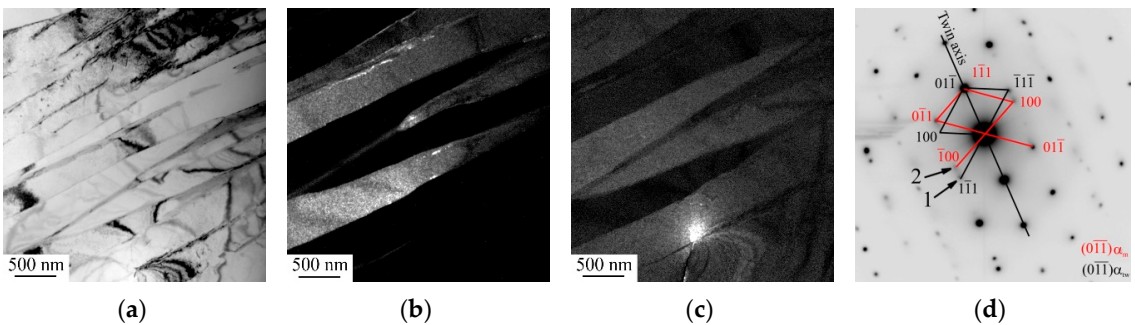

**Figure 10.** TEM bright- (**a**) and dark-field images (**b**,**c**) and associated SAED pattern (**d**) of the microstructures of the core of the EBF³-fabricated Ti–6Al–4V sample. **b**—dark-field TEM image obtained with the $1\bar{1}1_{\alpha\text{-Ti}}$ reflection (marked by an arrow 1 in (**d**)), **c**—dark-field TEM image obtained with the $\bar{1}00_{\alpha\text{-Ti}}$ reflection (marked by an arrow 2 in (**d**)).

It must be noted that the dissipation of the stored strain energy associated with martensitic transformations in the as-built Ti–6Al–4V parts under study occurs by the dislocation glide and deformation twinning.

X-ray diffraction in the Bragg–Brentano focusing geometry affords us the opportunity to reveal differences in the microstructure and phase composition of the outermost surface layer and the core region of EBF³ fabricated Ti–6Al–4V parts. It is found that the outermost surface layer of the as-built Ti–6Al–4V parts is represented by the hexagonal close-packed α titanium and the body-centered cubic β titanium. The evidence of the presence of the β-Ti phase within the outmost surface layer is an asymmetric broadening of the XRD peak at 2θ = 40.2° belonging to α-Ti (101) (Figure 11, curve 1). This particular peak shape can be considered as the superposition of several peaks for α-Ti (101) and β-Ti (110). The volume fraction of the β-Ti phase does not exceed 5%. Remarkably, as it is clearly seen from the relative intensities of the α-Ti diffraction peaks, the outermost surface layer of the as-built samples is characterized by a preferential crystallographic direction <101>. It is also worth noting a shift in the position of the α-Ti peaks towards lower angles, which is related to compressive stresses developed in the outermost surface layer. The measured compressive stresses value is about 700 MPa.

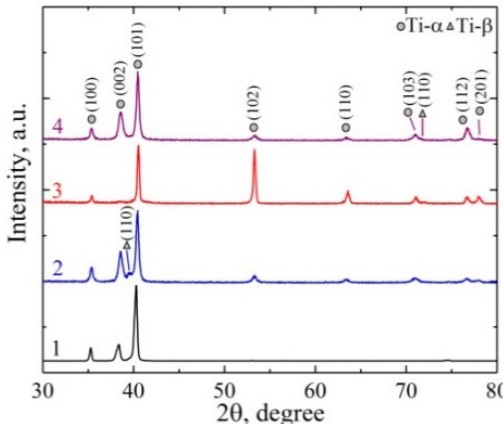

**Figure 11.** X-ray diffraction patterns of as-built (1—outermost surface layer, 2—core region) and irradiated Ti–6Al–4V sample (3—molten surface layer, 4—heat affected zone).

Significant changes in the X-ray diffraction patterns of as-built Ti–6Al–4V parts occur when their outermost surface layer has been removed. As seen from Figure 11 (curve 2), the relative intensity ratio of the α-Ti peaks changes towards that characteristic for untextured titanium polycrystals. Moreover, the separated X-ray diffraction peak of the β-Ti phase located at 2θ = 39.5° is clearly identified, revealing the increase in the β phase volume fraction up to 8%. The value of compressive

residual stresses developed in the core of the Ti–6Al–4V parts is equal to 600 MPa, which is a bit lower than the stresses in the outermost surface layer.

Nowadays, a number of models, including the temperature gradient mechanism model [28], cool-down phase model [29], etc., have been developed to explain the sign and value of residual stresses in metal parts produced by additive manufacturing. Both experimental and numerical investigations reveal that residual stresses typically tend to be compressive near the center of EBM and SLM parts and tensile near their surfaces. Besides, residual stresses in the parts produced by the EBM process are an order of magnitude lower compared with stresses in SLM [30]. Nevertheless, it is well-documented that preheating of a baseplate or powder bed can drastically decrease residual tensile stresses or even change their sign if the preheating temperature exceeds 570 °C [31,32]. Along with the reheating of the solidified metal close to the melt pool, some residual oxygen in the vacuum chamber as well as the strong affinity of titanium for oxygen is a more probable reason for the generation of high compressive stress within the outermost surface layer of the samples under study. Although EDS analysis does not show the presence of oxygen in the as-build Ti–6Al–4V parts, the high-magnified TEM images occasionally reveal $TiO_2$ (brookit) nanoparticles precipitated inside $\alpha$ lathes (Figure 7c). It can be expected that the interstitial (primarily oxygen) content is also responsible for the instability of the $\alpha$- and $\beta$-Ti crystal lattices, resulting in strain-induced martensite transformations within the outermost surface layer of the samples during the $EBF^3$ process.

Plan-view SEM-imaging of the irradiated Ti–6Al–4V sample makes a good case for the outermost surface layer melting during continuous electron beam treatment (Figure 12a,b). The boundaries of equiaxial grains with a size of about 20 to 30 µm are clearly observed on the grooved surface of the $EBF^3$-fabricated Ti–6Al–4V parts subjected to the treatment. The latter is due to a varied growth rate of prior $\beta$ grains depending on their crystal orientation. The grains are characterized by the rippled topography, confirming the rapid solidification of the molten surface layer into $\beta$-grains followed by the solid state $\beta \rightarrow \alpha'$ martensitic transformation. Cross-section metallographic observations also reveal the appearance of the recrystallized grains within the molten surface layer approximately 70 µm thick (Figure 12c). Notably, the grains grow in a rather equiaxial manner that indicates the lack of the dominant heat transfer path in the melt pool. Similar to [33], the cooling rate of the molten surface layer is estimated from the size of the prior $\beta$ grains that appeared during electron beam treatment. It is found that an approximate cooling rate reaches a value of $2.5 \times 10^5$ K/s, which is much higher than the typical cooling rate of the molten surface layer produced by continuous electron beam treatment ($10^3$–$10^4$ K/s) that was pointed out in [34,35]. It can be deduced that the ultimate cooling rate is due to the high efficiency heat sink by means of water cooling of the titanium baseplate. Nevertheless, new experiments are necessary to gain insight into the dependence of the cooling rate on the nature of irradiated materials and the processing parameters of electron beam treatment.

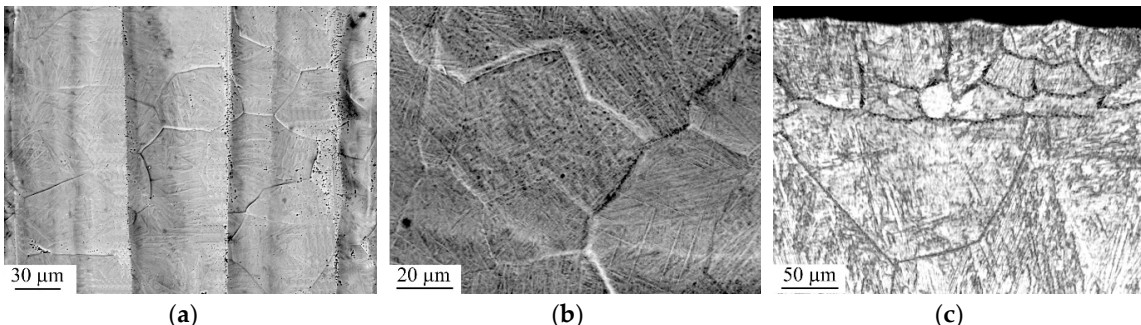

**Figure 12.** Plan-view (**a**,**b**) and cross-section (**c**) SEM-images of $EBF^3$-fabricated Ti–6Al–4V samples subjected to electron beam treatment.

The electron beam treatment significantly changes the intensities of the $\alpha$ phase XRD peaks as compared to the XRD pattern of the outermost surface layer of the as-built Ti–6Al–4V samples,

which provides evidence of its melting followed by crystallization (Figure 11, curves 3 and 1). It is clearly observed that the diffraction peak corresponding to the (002) crystal plane of α-Ti almost disappeared whereas the intensity of the peak for the (102) plane is drastically increased. In such a case, the treatment does not change the volume fraction of the β phase in the molten outermost surface layer, whose volume still does not exceed 2%. The latter is confirmed by the missing of separate XRD peaks belonging to β-Ti. In addition, since the electron beam treatment usually results in high tensile stresses developing in the molten surface layer of structural materials due to thermal contraction, it is no wonder that the compressive residual stresses in the melted surface layer of the irradiated Ti–6Al–4V parts decrease down to 200 MPa.

The cross-section TEM study indicates that the electron beam treatment transforms the outermost surface layer with coarse lath-shaped α-grains into the fine acicular martensitic α′ phase (Figure 13). Moreover, the fragmentation of the α′ phase by twinning occurs. The identified ultimate width of the twins is about 30 to 50 nm. The high cooling rate is also obviously responsible for the redistribution of V and Al within the molten surface layer. As a result, the vanadium enrichment in the α phase and the decrease of the vanadium content in the β phase occur, reducing the volume fraction of the β phase and forming the α″ orthorhombic phase, which is located at the twin boundaries and between the fine acicular α′ martensitic phases. It is interesting to mention at this point that the α″ phase is not usually observed in Ti–6Al–4V titanium alloy after pulse or continuous electron beam treatment due to fast (109 K/s) [36] or slow (103–104 K/s) [33] cooling rates of the molten surface layer, respectively. Meanwhile, solidification at an intermediate rate (e.g., 104–106 K/s), which is typical of the cooling rate during EBM or SLM processes [37], is responsible for the smaller vanadium enrichment in the β phase, resulting in the appearance of the metastable α″ orthorhombic phase in additively manufactured Ti–6Al–4V parts [22,23,38].

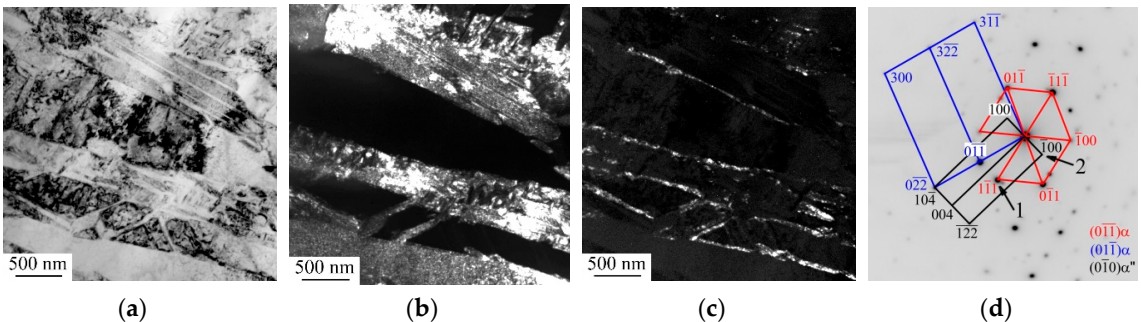

**Figure 13.** TEM bright- (**a**) and dark-field images (**b**,**c**) and associated SAED pattern (**d**) of the microstructures of the melted surface layer of the EBF³-fabricated Ti–6Al–4V sample subjected to electron beam treatment. **b**—dark-field TEM image obtained with the $1\bar{1}1_{α\text{-Ti}}$ reflection (marked by an arrow 1 in (**d**)). **c**—dark-field TEM image obtained with the $\bar{1}00_{α''\text{-Ti}}$ reflection (marked by an arrow 2 in (**d**)).

Beneath the molten surface layer, there is an approximately 500 μm thick deep layer, which exceeded the β transus temperature (so-called heat-affected zone) under continuous electron beam treatment. Due to fast cooling, the α → β → α phase transformation within prior β grains in the heat-affected zone is accompanied by the formation of supersaturated solid solution and its subsequent decomposition. The process is absolutely similar to the one developed in the solidified metal close to the melt pool during layer-by-layer deposition.

To perform an XRD analysis of the heat-affected zone, the 100 μm thick molten surface layer of the samples was removed by mechanical polishing. XRD patterns from the heat-affected zone and the core region of Ti–6Al–4V samples are clearly seen to be similar (Figure 11, curves 4 and 2). However, the volume fraction of the β phase within the heat-affected zone does not exceed 3%, which is lower than the volume fraction of the β-Ti phase in the core region. The asymmetric broadening of the XRD peak at 2θ = 71.7° belonging to β-Ti (110) only evidences the presence of the β-Ti phase within

the heat-affected zone. The value of the compressive residual stresses developed in the heat-affected zone of Ti–6Al–4V samples is equal to 500 MPa.

### 3.3. Microhardness

Figure 14 clearly highlights that the outermost surface layer of EBF³-fabricated Ti–6Al–4V parts in the as-built state exhibits a higher Vickers microhardness compared to their core. As it is seen from curve 1 of Figure 14, the microhardness exhibits a constant value (5600 MPa) at depths up to 100 μm from the surface, which corresponds to the thickness of the outermost surface layer composed of pockets of massive martensite. It is evident that strong fragmentation of the course lath-shaped α-grains within the outermost surface layer through the slip band propagation, resulting in strain-induced martensite transformations and nanosized β grains precipitation, is responsible for the surface hardening process. Beneath the outermost surface layer, the microhardness value tends to gradually decrease down to the core value (4500 MPa). This value is in good agreement with reported Vickers microhardness of Ti–6Al–4V parts additive manufactured via the EBM process (4.4–4.9 GPa) [39].

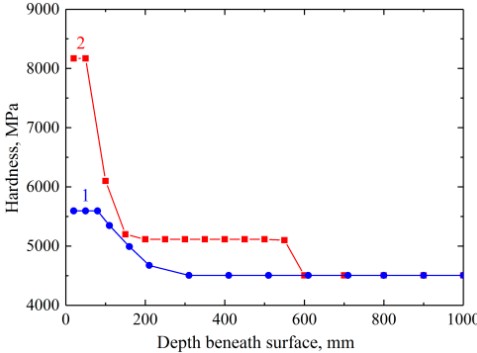

**Figure 14.** Distribution of Vickers microhardness from the outermost surface to the core of the EBF³-fabricated Ti–6Al–4V sample in the as-built state (1) and after continuous electron beam treatment (2).

The depth distribution of the Vickers microhardness for the EBF³-fabricated Ti–6Al–4V parts subjected to continuous electron beam treatment evidences the three-layer structure in the irradiated sample (Figure 14, curve 2). The molten surface layer, about 70 μm thick and composed of recrystallized equiaxial prior β grains with the fragmented acicular α′ phase, is characterized by the highest and constant value of the microhardness, exceeding 8200 MPa. Beneath the molten surface layer, the microhardness gradually decreases down to the value corresponding to the that of the heat affected zone (5200 MPa), whose microhardness value is also constant in the depth ranges from 150 to 550 μm below the surface of the irradiated sample. It is noted that the sharp interface between the molten surface layer and the heat affected zone caused a much higher gradient of microhardness beneath the molten surface layer (Figure 14, curve 2) in comparison with the microhardness gradient beneath the outermost surface layer in the as-built Ti–6Al–4V parts (Figure 14, curve 1). The total depth of the hardened surface layer, defined as a depth below the surface where the microhardness drops to nearly core value, reaches 600 μm.

### 3.4. Strain Response during Tensile Test

It is seen from curve 1 of Figure 15 that the ultimate tensile strength and plastic strain of EBF³-fabricated Ti–6Al–4V parts in the as-built state are equal to 770 MPa and 2%, correspondingly. As mentioned above, the samples for tensile tests were cut perpendicular to the build direction, i.e., along the surface finish in the form of horizontal layer bands and perpendicular to columnar prior β grains, respectively. Since the peak-trough profile roughness of the EBF³-fabricated sample is oriented perpendicular to their gauge length, the stress concentration effect of the peak–trough profile roughness is less pronounced. Nevertheless, as can be seen from comparison of the micrograph of the as-built

and mechanically polished Ti–6Al–4V parts stretched up to fracture (Figure 16a,b), vertical cracks are observed across the outermost surface layer, leading to non-crystallographic shear band initiation ahead of the cracks, whereas preliminary polishing removes the strain localization effect. It is also clearly visible from Figure 16a that the crack propagation is limited to a distance no greater than the thickness of the outermost surface layer (~100 µm), wherein shear bands oriented along conjugate directions of maximum shear stresses are propagated deeper into the core of the loaded parts. In doing so, shear band propagation significantly increases the strain energy stored in the defect structure of the loaded sample. Since the cracking promotes the intense strain energy release, it can be argued that surface cracking has a significant effect on the tensile properties of the Ti–6Al–4V parts in the as-built state.

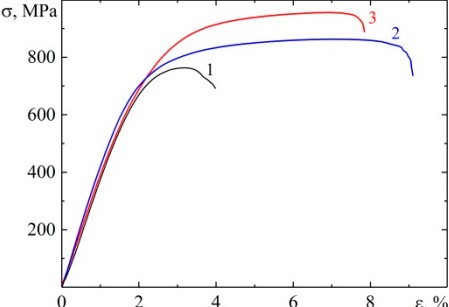

**Figure 15.** Tensile engineering stress–strain curves for as-built (curve 1), mechanically polished (curve 2), and irradiated Ti–6Al–4V samples (curve 3).

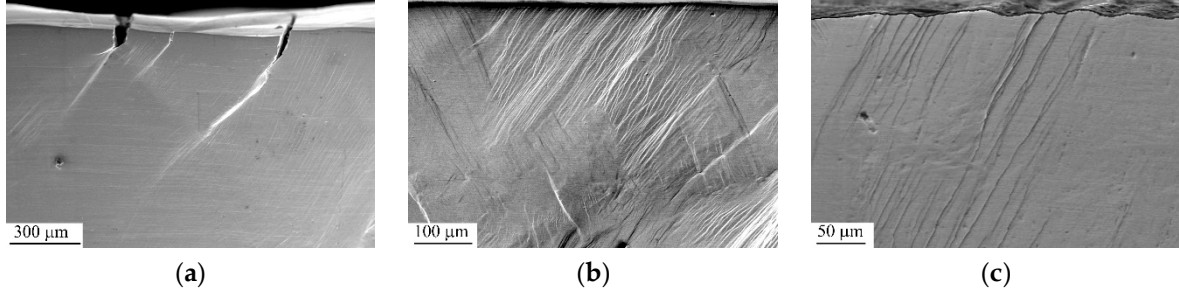

(**a**)                                        (**b**)                                        (**c**)

**Figure 16.** Cross-sectional SEM images of as-built (**a**), mechanically polished (**b**), or electron beam irradiated Ti–6Al–4V samples (**c**) subjected to uniaxial tension for $\varepsilon$ = 2% (**a**), 8% (**b**), and 6% (**c**).

The preliminary removal of a 1 mm thick surface layer by mechanical polishing withdraws the cracking and strain localization, whereas dislocation glide becomes the preferred mechanism of plastic deformation (Figure 16b). As a result, strength and ductility of the EBF$^3$-fabricated Ti–6Al–4V parts drastically increase (Figure 15, curve 2). Nevertheless, the values of the yield stress ($\sigma_y$) and the ultimate tensile strength ($\sigma_B$) of the mechanically polished samples under study are still lower than those of wrought alloys ($\sigma_y$ = 850 MPa; $\sigma_B$ = 930 MPa) given in the ASM (American Society for Metals) Handbook [40] due to the coarse columnar grains of the as-built Ti–6Al–4V sample.

Finally, electron beam treatment more significantly affects the strain response of the as-built Ti–6Al–4V parts under uniaxial tension (Figure 15, curve 3). The formation of equiaxial prior β grains of 20 to 30 µm in size with the fragmented acicular α' phase within the molten surface layer and decomposition of the supersaturated solid solution within the heat-affected zone of the irradiated Ti–6Al–4V sample increase its tensile strength up to 950 MPa wherein its plastic strain slightly decreases down to 6%. This shows itself as retardation of the nucleation of new dislocations and of the motion of the pre-existing dislocations both inside the molten surface layer and heat-affected zone of the loaded sample. However, since the movement of dislocations is suppressed incompletely and crack initiation

does not occur in the hardened surface layer (Figure 16c), the increase in strength of the irradiated Ti–6Al–4V samples is not accompanied by a significant reduction in their plasticity.

## 4. Conclusions

This paper summarized the results obtained when studying the effect of continuous electron beam treatment on the surface finish, microstructure, and mechanical properties of EBF³-fabricated Ti–6Al–4V parts. It was established that the as-built Ti–6Al–4V part is characterized by a surface finish in the form of horizontal layer bands, which are governed by the flow direction of the molten pool during the layer additive process. Although the continuous electron beam treatment resulted in a grooved surface topography of the irradiated EBF³-parts, their melted surface layer became significantly smoother in comparison with the as-build parts. The height differences between the peak and valley of the profiles directed parallel to the build direction thrice decreased (from 150 to 50 μm) upon the treatment.

The microstructure of the approximately 100 μm thick outermost surface layer of the as-built samples was composed of massive martensite, which consisted of coarse α lathes (100 μm in length and 5 to 10 μm in thickness). Moreover, the precipitation of the β phase at the high-angle boundary of secondary α lathes occurred due to the formation of supersaturated solid solution and its subsequent decomposition in the solidified metal close to the melt pool during layer-by-layer deposition. The dissipation of the stored strain energy (related to martensitic transformations) resulted in slip band propagation within the outermost surface layer. The latter promotes both the lath fragmentation and the martensitic transition of a highly strained metastable β-Ti crystal lattice into the lattice of an orthorhombic martensite α″ phase. Continuous electron beam post-treatment of the EBF³-fabricated Ti–6Al–4V parts caused the formation of equiaxial prior β grains of 20 to 30 μm in size with the fragmented acicular α′ phase within the molten surface layer approximately 70 μm thick and decomposition of supersaturated solid solution within the heat-affected zone of approximately 500 μm thick.

Fragmentation of massive martensite as well as precipitation of nanosized β grains within the outermost surface layer were responsible for their enhanced Vickers microhardness (5600 MPa) compared to the core of the as-built Ti–6Al–4V part (4500 MPa). However, vertical cracks across the outermost surface layer, leading to non-crystallographic shear band initiation ahead of cracks, were observed under tension, resulting in low values of ultimate tensile strength (770 MPa) and plastic strain (2%) of the EBF³-fabricated Ti–6Al–4V parts in the as-built state. Subsequent continuous electron beam treatment re-melted their brittle outermost surface layer and drastically changed their microstructure, causing an increase in the surface microhardness (8200 MPa) and tensile strength (950 MPa) of the samples under study.

**Author Contributions:** A.P. (Alexey Panin) analyzed the results, and prepared the paper, M.K. carried out surface finish measurements, performed SEM investigations, analyzed the results, and prepared the paper. O.P. performed TEM investigations and analyzed the results. E.S. carried out microhardness and tension testing and analyzed the results. A.P. (Alexandra Panina) performed X-ray diffraction studies and analyzed the results. S.M. carried out sample fabrication by the EBF³ method and performed continuous electron beam post-treatment.

**Funding:** This research was funded by the RUSSIAN SCIENCE FOUNDATION, grant number 18-19-00559.

**Conflicts of Interest:** The authors declare no conflict of interest.

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
