# Peer review of "Continuous Electron Beam Post-Treatment of EBF3-Fabricated Ti–6Al–4V Parts"

_metals, doi:10.3390/met9060699_

Round 1
Reviewer 1 Report
Dear Dr. Panin,
This manuscript addresses the microstructure of EBF3-fabricated Ti-6Al-4V and the effect of electron beam post-treatment. This manuscript needs revision to be accepted.
Please sow the detail of the experimental method about the electron beam post-treatment. For example, please show not only beam condition but also the configuration of experimental setup.
Please show how to prepare TEM samples.
In lines 275-276, you mentioned "the high-magnified TEM images occasional reveal TiO2 (brookit) nanoparticles precipitated inside alpa lathes." Please show the TEM image which reveal TiO2.
Sincerely yours.
Author Response
Thank you for your useful comments and input to this paper. Responses to each issue are provided below. We also made some minor changes in the manuscript according to most of the comments and suggestions of other reviewers. All the changes were marked in red in the revised manuscript.
Please sow the detail of the experimental method about the electron beam post-treatment. For example, please show not only beam condition but also the configuration of experimental setup.
Response:
The configuration of experimental setup for electron beam post-treatment was inserted into the manuscript as figure 1.
Please show how to prepare TEM samples.
Response:
The procedure of TEM sample preparation was also inserted into the manuscript.
In lines 275-276, you mentioned "the high-magnified TEM images occasional reveal TiO2 (brookit) nanoparticles precipitated inside alpa lathes." Please show the TEM image which reveal TiO2.
Response:
High resolution TEM bright- and dark-field images, revealing the presence of TiO2 nanoparticles were embedded within the manuscript.
Reviewer 2 Report
In this manuscript, the authors studied continuous electron beam post-treatment of Ti-6-4.
I have comment in Fig. 3. Please consider to re-etch the samples and replace Fig. 3 with higher quality images.
Author Response
We appreciate reviewer very much for the kind comments and recommendations of our manuscript. We have replaced Fig. 3 with higher quality images. Thank you for your patience and time. We also made some minor changes in the manuscript according to most of the comments and suggestions of other reviewers. All the modifications were marked in red in the revised manuscript.
Reviewer 3 Report
The article claims to study the effect of continuous electron beam treatment on the surface finish, microstructure and mechanical properties of EBF3-fabricated Ti-6Al-4V parts.
The manuscript is written on a good level, with high scientific erudition of the authors. I have found there all the necessary information regarding the research, from the theoretical basement, through the research description, up to the results and their evaluation.
I have no additional comments to the article, so I recommend publishing the article as it is.
Author Response
We sincerely appreciate the reviewer for his patience and guidance. Nevertheless we have slightly modified the manuscript according to most of the comments and suggestions of other reviewers. All the modifications were marked in red in the revised manuscript.
Reviewer 4 Report
Introduction: ‘low-coast building’ -> ‘low-cost building’
Authors attribute the coarse grain structure to the repeated heating, but this is also true of other AM processe with a finer structure. The role of the traveling speed and melt pool size in the coarse grain structure should be addressed in comparison to laser powder bed fusion methods.
Page 3: Line 125-126: Surface roughness features are attributed to the surface tension of the melt pool with citation or other evidence. It is not obvious why the features of Figure 1(b) would be caused by surface tension.
Line 152-153: What is the “alternative layer thickness” referred to here?
Figure 4: Does not reference part c of the figure in the caption.
-
Author Response
We greatly appreciate the reviewer’s time and effort for carefully reading our manuscript, helpful suggestions and comments. Modifications have been made accordingly, which were marked red in the revised manuscript. The followings are our point-to-point responses and modification actions (colored) to the reviewer’s comments.
Introduction: ‘low-coast building’ -> ‘low-cost building’
Response:
The grammatical mistakes have been corrected.
Authors attribute the coarse grain structure to the repeated heating, but this is also true of other AM processe with a finer structure. The role of the traveling speed and melt pool size in the coarse grain structure should be addressed in comparison to laser powder bed fusion methods.
Response:
Comparative analysis of the width and depth of the molten pool formed in Ti-6Al-4V parts during SLM, EBM and EBF3 process was embedded within the manuscript.
Page 3: Line 125-126: Surface roughness features are attributed to the surface tension of the melt pool with citation or other evidence. It is not obvious why the features of Figure 1(b) would be caused by surface tension.
Response:
It is well-documented that under the effect of surface tension, the molten pool takes the shape of a circular. Fragmentation of the remelted tracks is a well-known drawback of additive manufacturing processes referred to as the “balling” effect. When the electron beam moves, there is a temperature gradient between the electron beam and the solidifying zone; this generates a shear force on the liquid surface that is contrasted by surface tension forces [Ramos JA, Bourell DL, Beaman JJ (2003) Surface over-melt during laser polishing of indirect-SLS metal parts. Mater Res Soc Symp 758:53-61]. A higher temperature of the melt promotes better surface tension and wettability, which is beneficial to the formation of a stable melt pool and smooth surface finish [Gu, D.; Shen, Y. Balling phenomena in direct laser sintering of stainless steel powder: Metallurgical mechanisms and control methods. Mater. Des. 2009, 30, 2903–2910]. The features of the tracks’ instability depend of electron beam power, scanning speed, powder layer thickness, substrate material, physical properties of the feedstock, etc. Physical mechanism of surface roughening of parts produced by additive manufacturing is described in Ref. [Yadroitsev, I.; Gusarov, A.; Yadroitsava, I.; Smurov, I. Single track formation in selective laser melting of metal powders // J. Mater. Process. Technol. 2010, 210, 1624–1631.]
Line 152-153: What is the “alternative layer thickness” referred to here?
Response:
To clarify, the word “alternative” has been changed to word “alternating”.
Figure 4: Does not reference part c of the figure in the caption.
Response:
Thank you for your very careful review of our manuscript. The mistake has been corrected.